# A circuit mechanism for the propagation of waves of muscle contraction in *Drosophila*

**Akira Fushiki[1,2], Maarten F Zwart[2,3], Hiroshi Kohsaka[1], Richard D Fetter[2], Albert Cardona[2]\*, Akinao Nose[1,4]\***

[1]Department of Complexity Science and Engineering, Graduate School of Frontier Sciences, University of Tokyo, Tokyo, Japan; [2]Janelia Research Campus, Howard Hughes Medical Institute, Ashburn, United States; [3]Department of Zoology, University of Cambridge, Cambridge, United Kingdom; [4]Department of Physics, Graduate School of Science, University of Tokyo, Tokyo, Japan

**Abstract** Animals move by adaptively coordinating the sequential activation of muscles. The circuit mechanisms underlying coordinated locomotion are poorly understood. Here, we report on a novel circuit for the propagation of waves of muscle contraction, using the peristaltic locomotion of *Drosophila* larvae as a model system. We found an intersegmental chain of synaptically connected neurons, alternating excitatory and inhibitory, necessary for wave propagation and active in phase with the wave. The excitatory neurons (A27h) are premotor and necessary only for forward locomotion, and are modulated by stretch receptors and descending inputs. The inhibitory neurons (GDL) are necessary for both forward and backward locomotion, suggestive of different yet coupled central pattern generators, and its inhibition is necessary for wave propagation. The circuit structure and functional imaging indicated that the commands to contract one segment promote the relaxation of the next segment, revealing a mechanism for wave propagation in peristaltic locomotion.

**\*For correspondence:** cardonaa@ janelia.hhmi.org (AC); nose@k.u-tokyo.ac.jp (AN)

**Competing interests:** The authors declare that no competing interests exist.

## Introduction

Animal locomotion is generated by coordinated activation of muscles throughout the body (*Grillner, 2003*; *Marder and Calabrese, 1996*; *Mulloney et al., 1998*). For example, during axial locomotion such as lamprey swimming and *Drosophila* larval crawling, muscles present in each segment are sequentially activated along the body axis in a stereotypic temporal and spatial pattern (*Grillner, 2003*). How neural networks, including those underlying central pattern generators (CPGs) and sensory feedback circuits, orchestrate the precisely timed activation of motor and premotor neurons in multiple body segments remains poorly understood.

Previous studies have identified functional connectivity among neurons that are important for rhythmic movements and intersegmental coordination, using electrophysiology in leech (*Kristan et al., 2005*), lamprey (*Grillner, 2003*) and crayfish (*Smarandache-Wellmann and Gratsch, 2014*; *Smarandache-Wellmann et al., 2014*; *Smarandache et al., 2009*) among others. Recent studies in mouse (*Goetz et al., 2015*; *Talpalar et al., 2013*), zebrafish (*Kimura et al., 2013*) and worm (*Wen et al., 2012*) revealed the roles played by different classes of interneurons in the regulation of motor coordination. A complete wiring diagram with synaptic resolution of motor circuits spanning the entire nervous system would contextualize current knowledge and facilitate advancing our understanding of motor pattern generation.

**eLife digest** Rhythmic movements such as walking and swimming require the coordinated contraction of many different muscles. Throughout the animal kingdom, from insects to mammals, animals possess specialized circuits of neurons that are responsible for producing these patterns of muscle contraction. These circuits are known as 'central pattern generators'.

Central pattern generators are made up of multiple types of neurons that exchange information. However, it is unclear how neurons controlling the movement of one part of the body relay information to neurons controlling the movement of other parts. To answer this question, Fushiki et al. used larvae from the fruit fly *Drosophila melanogaster* as a model, and combined techniques such as electrophysiology and electron microscopy with measures of the insect's behavior.

Fruit fly larvae have bodies that are made of segments, and they can contract and relax these segments in a sequence to propel themselves forwards or backwards. The contraction of one segment is accompanied by relaxation of the segment immediately in front. Fushiki et al. found that each body segment contains a copy of the same basic neuronal circuit. This circuit is made up of excitatory and inhibitory neurons. Both types of neurons regulate movement, but the inhibitory neurons must be suppressed for movement to occur.

The experiments also showed that each circuit receives both long-range input from the brain and local sensory feedback. This combination of inputs ensures that the segments contract and relax in the correct order. Future challenges are to determine how the brain controls larval movement via its long-range projections to the body. A key step will be to map these circuits at the level of the individual neurons and the connections between them.

Larval *Drosophila* has recently emerged as a powerful model system for studying the neural regulation of locomotion (*Heckscher et al., 2012*; *Kohsaka et al., 2014*; *Landgraf et al., 1997*). Its primary locomotor pattern consists of wave-like muscular contractions that propagate either from posterior to anterior segments (forward movement) or from anterior to posterior (backward movement) segments (*Heckscher et al., 2012*). This sequential activation of segmental musculature is generated by segmentally interconnected circuits in the ventral nerve cord (VNC). The basic pattern of motor activity can be observed as fictive locomotion in dissected larvae or in isolated nerve cords, to which localized optogenetic manipulation can be applied (*Fox et al., 2006*; *Kohsaka et al., 2014*; *Pulver et al., 2015*). Furthermore, the larva also is capable of a variety of other locomotive patterns and can adjust to changes in environmental conditions (*Godoy-Herrera, 1994*; *Hwang et al., 2007*; *Ohyama et al., 2015*; *Vogelstein et al., 2014*). Powerful genetic tools, including a resource of GAL4 drivers (*Pfeiffer et al., 2010*), allow for the manipulation of the activity of uniquely identified neurons in this simple nervous system (*Li et al., 2014*; *Manning et al., 2012*). These genetic tools enable optogenetic manipulation and the monitoring of neural activity in larvae in the context of mapped circuitry thanks to novel circuit mapping tools (*Saalfeld et al., 2009*) and an electron microscopy volume of the complete central nervous system of the larva (*Ohyama et al., 2015*).

Here, we report a novel circuit and mechanism for mediating wave propagation in peristaltic locomotion. We screened GAL4 driver lines and identified neurons that are active with the peristaltic wave of larval muscle contraction. We then mapped the circuits with synaptic resolution in which these neurons are embedded, and we found a repeating modular circuit formed by an inhibitory (GDL) and an excitatory neuron (A27h) in each hemisegment, connected in a chain across consecutive segments. Using optogenetics and functional imaging, we determined that the inhibitory neuron GDL is necessary for both forward and backward locomotion, but the excitatory neuron A27h is necessary only for forward locomotion, suggesting underlying coupled circuits. Body-wide activation of GDL led to the paralysis of the abdominal segments, but its localized activation in a few consecutive segments was sufficient to arrest the wave of propagation. Taken together, our findings define a mechanism for wave propagation in which the contraction of one segment is concomitant with the relaxation of the adjacent anterior segment, and the cessation of contraction is in turn coupled with the stimulation of contraction in next anterior segment. The logic of this network allows for

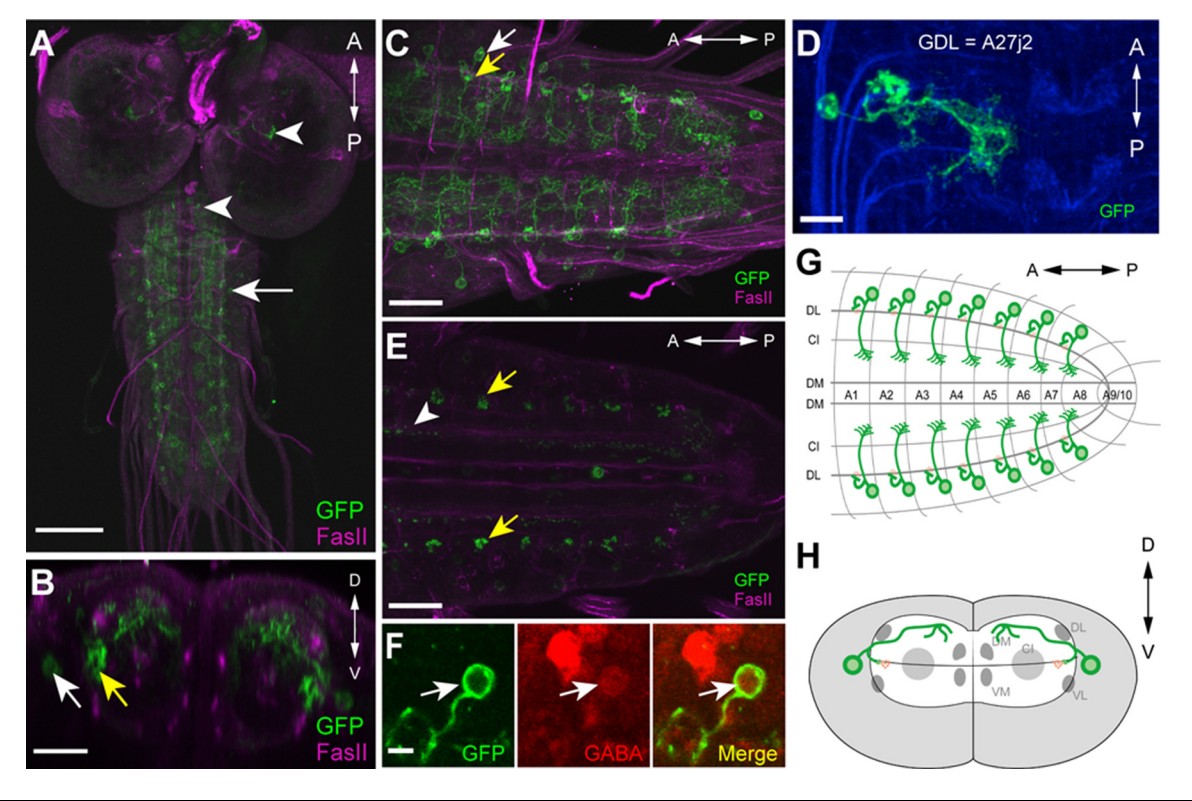

**Figure 1.** Morphology of GDLs. The *GDL-GAL4* (*9-20-GAL4, iav-GAL80*) drives expression in GDLs and a small number of cells in the brain and SEZ. All panels show dissected third instar larval CNS. (A–C) Morphology of GDLs was visualized with *10xUAS-IVS-myr::GFP* reporter expressed by *GDL-GAL4*. Anti-GFP (green) and anti-FasII (magenta) antibodies were used. (A) A low magnification view showing *GDL-GAL4* expression in a GDL (arrow) and in a small number of cells in the brain and SEZ (arrowheads). (B) A cross sectional view of an abdominal segment. White arrow denotes the cell body of a GDL in a dorsolateral area of the VNC. Yellow arrow denotes the presynaptic terminals of a GDL. (C) A dorsal view showing segmentally repeated GDLs in the VNC. Each GDL extends its neurites locally within the segment. Anterior is to the left and posterior is to the right. (D) An image of a fluorescently labelled single-cell clone of GDL (courtesy of James W. Truman, HHMI Janelia Research Campus). GDL is also annotated as A27j2. (E) *UAS-syt::GFP* was used as a reporter to visualize presynaptic terminals of GDLs (yellow arrows). Signals seen in a medial region (arrowhead) are likely presynaptic terminals of descending neurons from the brain or SEZ (*Figure 7—figure supplement 1B*). (F) The cell body of GDLs was positive for GABA. (G, H) Schematic drawings of GDLs. Scale bar represents 80 μm in (A), 30 μm in (C, E), 20 μm in (B), 10 μm in (D) and 5 μm in (F). (See also *Figure 1—figure supplement 1*.)

The following figure supplement is available for figure 1:

**Figure supplement 1.** Expression driven by *9-20-GAL4* and *inactive (iav)-GAL80*.

additional models for coordinated muscle contraction that incorporate feedback from stretch receptors and also descending neurons from the subesophageal zone (SEZ).

## Results

### GDLs are pairs of segmentally repeated GABAergic interneurons

To identify interneurons that are involved in the regulation of larval locomotion, we screened for interneurons that exhibit an activity pattern correlated with larval locomotion. In previous studies, we reported on two classes of putative premotor interneurons (PMSIs and GVLIs) that inhibit motor neurons via glutamatergic transmission (*Itakura et al., 2015*; *Kohsaka et al., 2014*). In this study, we selected for GABA-positive and rhythmically active neurons within sparsely expressing GAL4 lines and identified a class of interneurons, which we named GABAergic dorsolateral neurons (GDLs, also annotated as A27j2).

GDLs are a pair of neurons present in each abdominal segment, and were identified in *9-20-GAL4* (*Hughes and Thomas, 2007*). This line drives expression not only in GDLs but also in a subset of mechanoreceptors (the chordotonals) and a small number of cells in the brain and SEZ (*Figure 1* and *Figure 1—figure supplement 1A–C*). Since expression in the mechanoreceptors would complicate anatomical and functional analyses of GDLs, we used the promoter of the *inactive (iav)* gene, which is known to be specifically expressed in the mechanoreceptors (*Kwon et al., 2010*), to generate *iav-GAL80* (see Materials and methods). When combined with *9-20-GAL4*, *iav-GAL80* suppressed the GAL4-driven expression in the mechanoreceptors without affecting the expression in GDLs (*Figure 1—figure supplement 1D,E*). We used the combined line (*9-20-GAL4, iav-GAL80*; hereafter referred to as *GDL-GAL4*) as a driver for GDLs in the following experiments.

We studied the morphology of GDLs with *GDL-GAL4* driving the expression of myr-GFP (*Pfeiffer et al., 2010*). GDLs project their axons to a lateral area of the neuropile under the DL tract (FasII landmark system; [*Landgraf et al., 2003*]) and present their dendrites in the motor domain (*Figure 1A–C*). Clonal analyses further confirmed the projection pattern (*Figure 1D*). We labeled the axon terminals with the presynaptic marker syt::GFP (*Figure 1E*) and also determined GDLs as GABAergic with antibody staining (*Figure 1F*). To summarize, GDLs are segmental pairs of GABAergic interneurons local to each segment in the abdominal VNC (*Figure 1G,H*).

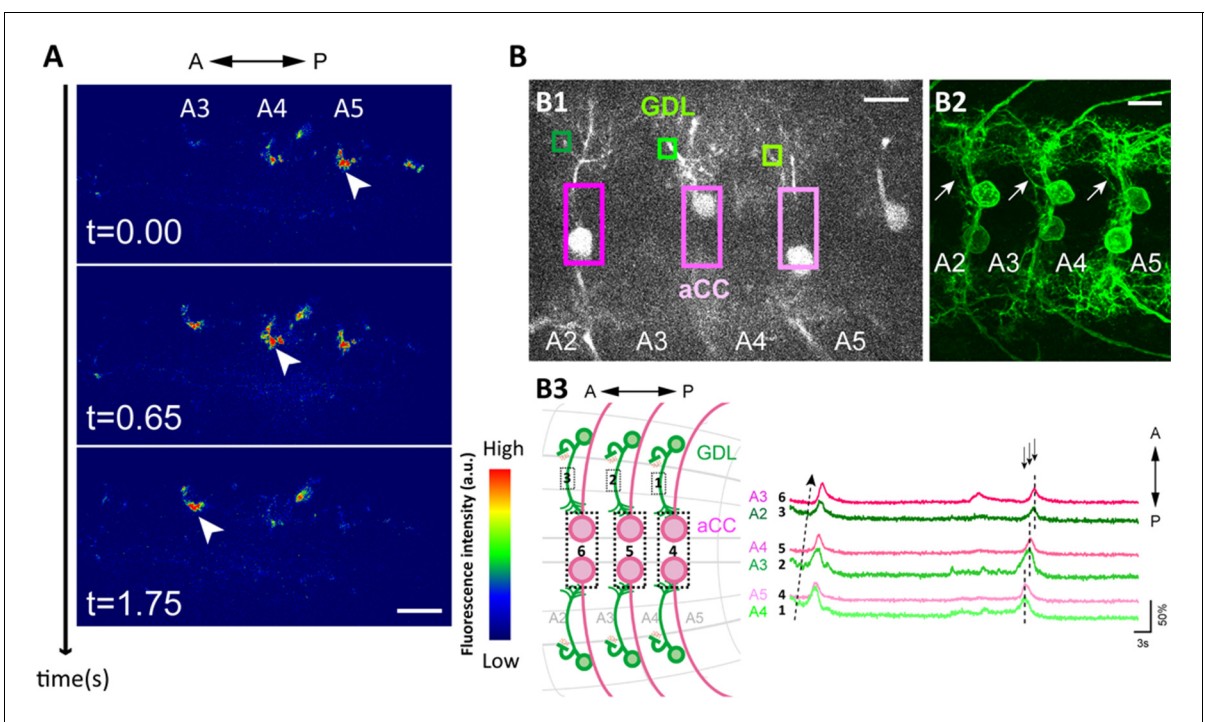

**Figure 2.** Wave-like activities of GDLs and their phase difference to motor neurons. (**A**) High-resolution calcium imaging of GDL activity in an isolated CNS preparation (*GDL-GAL4>20xUAS-IVS-GCaMP6m*). The increase in the calcium signal in the presynaptic terminals of GDLs propagated from posterior to anterior segments (arrowheads). (**B**) (B1) Regions of interest (ROI) used for the simultaneous calcium imaging of GDLs and aCC motor neurons. We compared the activities between the cell bodies of aCC motor neurons and the dendrites of GDLs (*GDL-GAL4,eve-GAL4>20xUAS-IVS-GCaMP6m*). (B2) Dendrites of GDLs (arrows) can be clearly distinguished from the neurites and cell bodies of aCC motor neurons (*GDL-GAL4,eve-GAL4>10xUAS-IVS-myr::GFP*). (B3) Temporal correlation between the activity of GDLs and aCC motor neurons. Note that activation of GDLs (green) occurs at a similar timing as that of aCC motor neurons in the next posterior segment (arrows, n = 10). Scale bar represents 15 µm in (**A**, **B**). (See also *Figure 2—figure supplement 1*.)

The following figure supplement is available for figure 2:

**Figure supplement 1.** Simultaneous imaging of GDLs activity and peristaltic waves.

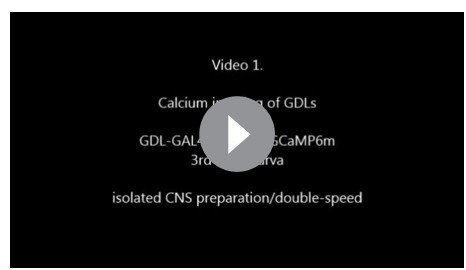

**Video 1.** Calcium imaging of GDLs. GCaMP6m was expressed in GDLs (*GDL-GAL4>20xUAS-GCaMP6m*). An isolated CNS preparation or semi-intact preparation from third instar larva. Double-speed or Quad-speed. (Related to *Figure 2*).

## GDLs show wave-like activities that propagate earlier than motor neurons

The isolated central nervous system (CNS) presents fictive, rhythmic motor patterns, which facilitates experimentation (*Fox et al., 2006*; *Pulver et al., 2015*). We monitored the activity of GDLs during fictive motor patterns of the dissected CNS by the targeted expression of GCaMP6m (*Chen et al., 2013*). We observed bilaterally symmetric propagation of calcium signals that travel along the segments both in forward and backward directions (*Pulver et al., 2015*) (*Figure 2A* and *Video 1*). We validated these observations with GCaMP6m imaging in a semi-intact preparation where we observed that GDLs are active simultaneously with muscle contractions in the adjacent segments (*Figure 2—figure supplement 1* and *Video 1*).

To further examine the coordinated activity of GDLs and muscles, we imaged the activity of GDLs and anterior corner cell (aCC) motor neurons (with *eve-GAL4*, [*Fujioka et al., 2003*]). We performed calcium imaging focusing on the dorsomedial region of the VNC where the dendrite of GDLs and the cell bodies of aCCs can be uniquely identified in the same focal plane (*Figure 2B1, 2*). We found GDLs in each segment were activated earlier than aCCs in the same segment and at a similar time as aCCs in the next posterior segment (*Figure 2B3* and *Video 2*). Thus, the activity of GDLs propagates along the segments ahead of the wave of motor neuron activity during forward locomotion. This suggests a role for GDLs in relaxing and resetting anterior segments prior to the arrival of the contraction wave.

## Synaptic transmission by GDLs is required for normal larval locomotion

To address the role of GDLs in larval locomotion, we first disrupted synaptic transmission in GDLs with *GDL-GAL4* driving the expression of tetanus toxin light chain (TNT) (*Sweeney et al., 1995*). We observed a significant decrease in the speed of larval locomotion (~30% slower than control, p<0.001; *Figure 3A,B* and *Video 3*). We also found a significant increase in the wave duration (~40% longer than control, p<0.001; *Figure 3C*) and a decrease in the number of forward peristaltic waves (~1/5th the normal frequency, *Figure 3D*). The *GDL-GAL4* is also expressed in a small subset of cells in the brain and SEZ. To test whether inhibition of GDLs alone is responsible for the observed TNT phenotype, we suppressed GAL4 activity in the VNC with *tsh-GAL80* (*Clyne and Miesenbock, 2008*). We analyzed the resulting expression pattern by combining *GDL-GAL4 and tsh-GAL80* and did not observe GAL4 activity in GDLs, however expression in the brain and SEZ remained intact (*Figure 7—figure supplement 1A,B*). The TNT phenotype was rescued with *tsh-GAL80*, indicating that GDLs, the only *GDL-GAL4*–expressing neurons in the VNC, were solely responsible for the phenotype (p<0.001; *Figure 3C*). As further proof, we specifically disrupted synaptic transmission in GDLs by disrupting GABA synthesis with RNAi directed against the *Glutamic acid decarboxylase 1 (Gad1)* gene, since other neurons in the *GDL-GAL4* pattern are not GABAergic (data not shown). RNAi knock-down of *Gad1* using two independent constructs that target different portions of the *Gad1* mRNA resulted in a similar increase in wave duration as we observed for TNT-expressing larvae (~40% longer than control, p<0.001;

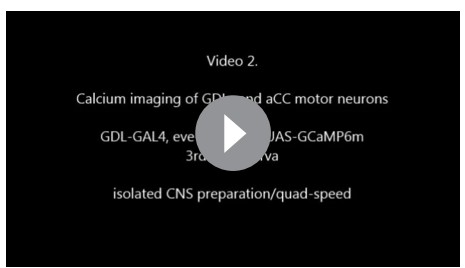

**Video 2.** Simultaneous calcium imaging of GDLs and aCC motor neurons. GCaMP6m was expressed in GDLs and subsets of motor neurons (*GDL-GAL4, eve-GAL4>20xUAS-GCaMP6m*). An isolated CNS preparation from third instar larva. Quad-speed. (Related to *Figure 2*).

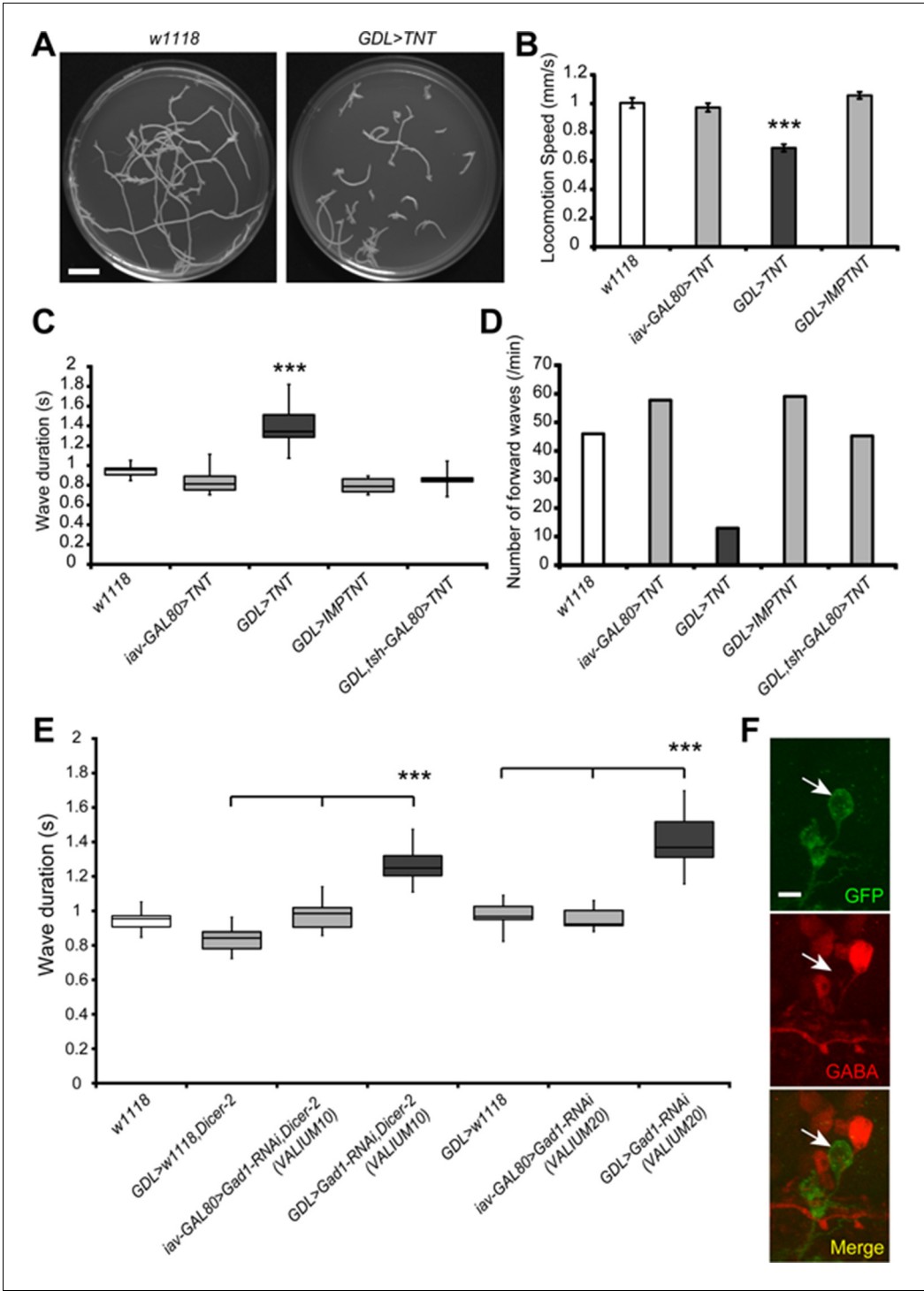

**Figure 3.** Inhibition of GDLs transmission reduced the speed and frequency of larval peristalsis. (**A**) The path taken by third instar larvae undergoing locomotion for 3 min is shown (left: $w^{1118}$, right: *GDL-GAL4>UAS-TNT*). (**B**) Inhibition of GDLs with *TNT* decreased the speed of larval locomotion (Locomotion speed, 0.69 ± 0.03 mm/sec [*GDL-GAL4>UAS-TNT*] compared to 1.00 ± 0.04 mm/sec [$w^{1118}$], 0.97 ± 0.03 mm/sec [*iav-GAL80>UAS-TNT*] and 1.06 ± 0.02 mm/sec [*GDL-GAL4>UAS-IMP(imperfect)TNT*]; p<0.001). (**C**) Expression of *TNT* in *GDL-GAL4* greatly increased the wave duration and the phenotype was rescued by *tsh-GAL80* (Wave duration, 1.40 ± 0.23 sec [*GDL-GAL4>UAS-TNT*] compared to 0.95 ± 0.08 sec [$w^{1118}$], 0.84 ± 0.12 sec [*iav-GAL80>UAS-TNT*], 0.80 ± 0.07 sec [*GDL-GAL4>UAS-IMP(imperfect)TNT*] and 0.90 ± 0.14 sec [*GDL-GAL4>tsh-GAL80,UAS-TNT*]; p<0.001). (**D**) *TNT*-mediated inhibition also caused a significant decrease in the frequency of larval locomotion (Number of forward

*Figure 3 continued*

waves, 13.0 waves/min [*GDL-GAL4>UAS-TNT*] compared to 46.0 waves/min [*w1118*], 57.8 waves/min [*iav-GAL80>UAS-TNT*], 59.1 waves/min [*GDL-GAL4>UAS-IMP(imperfect)TNT*] and 45.3 waves/min [*GDL-GAL4,tsh-GAL80>UAS-TNT*]). (**E**) Expression of two independent *Gad1-RNAi* transgenes in GDLs also increased the wave duration (Wave duration, 1.27 ± 0.1 sec [*GDL-GAL4>Gad1-RNAi(VALIUM10),Dicer-2*] compared to 0.84 ± 0.08 sec [*GDL-GAL4>w1118,Dicer-2*] and 0.98 ± 0.09 sec [*iav-GAL80>Gad1-RNAi(VALIUM10),Dicer-2*], 1.41 ± 0.05 sec [*GDL-GAL4>Gad1-RNAi(VALIUM20)*] compared to 0.98 ± 0.03 sec [*GDL-GAL4>w1118*] and 0.96 ± 0.02 sec [*iav-GAL80>Gad1-RNAi(VALIUM20)*]; p<0.001). (**F**) *GDL-GAL4;10xUAS-IVS-myr::GFP* driving *Gad1-RNAi(VALIUM20)* showed a significant reduction of GABA immunoreactivity of GDLs. Box plots in (**C** and **E**) indicate the median value (horizontal line inside the box), 25–75% quartiles (box), and the data range (whiskers). Statistical significance was determined by Student *t*-test or one-way ANOVA followed by Tukey's test for multiple comparisons (***p<0.001). For all conditions in each figure, n = 20 in (**B**) and n = 10 in (**C**, **D**, **E**). Scale bar represents 15 mm in (**A**) and 5 µm in (**F**).

*Figure 3E,F* and *Video 3*). These results show that the activity of GDLs is required for larvae to crawl at a normal speed and for normal muscle contraction wave frequency. Therefore, GABAergic transmission is critical for the function of GDLs and larval locomotion.

## A neural circuit for coordinating wave propagation

Having identified GDLs as necessary for propagating peristaltic waves, we then studied the neural circuit basis for GDL function. First, we determined that GDLs do not synapse directly onto motor neurons by using GRASP (*Feinberg et al., 2008*; *Gordon and Scott, 2009*), expressing each half of the GFP protein in GDLs and motor neurons, respectively (*Figure 4—figure supplement 1*). To confirm this, we then identified GDLs in an electron microscopy (EM) volume comprising the entire larval CNS (*Figure 4A*) and reconstructed all neurons synaptically connected to GDLs in segment A1, none of which were motor neurons (*Figure 4—figure supplement 2*, *3*). We also found that no strongly connected GDL partners synapse with each other, suggesting that GDLs act as hub neurons, with the potential to orchestrate activity patterns of postsynaptic neurons (*Figure 4B*). One of the top synaptic GDL partner cell types (by number of synapses), connected both presynaptically ("upstream") and postsynaptically ("downstream"), is the segmentally repeated premotor interneuron A27h (*Figure 4C,D* and *Figure 5—figure supplement 1A*). Interestingly, though all the downstream premotor interneurons were found in the same segment as GDLs, all the upstream premotor interneurons were located in the next posterior segment (*Figure 4D*). Furthermore, GDLs receive the inputs from somatosensory neurons (vdaA and vdaC class II dendritic arborization neurons; *Figure 4D*) that likely mediate gentle touch (*Tsubouchi et al., 2012*). Taken together, this arrangement configures a feed-forward circuit in which premotor interneurons of one segment not only drive motor neurons in the same segment but also transmit an inhibitory signal to their own homologs in the adjacent anterior segment via GDLs (*Figure 4E*), in parallel with a synaptic pathway for sensory feedback that also regulates transmission of the peristaltic wave (see Discussion).

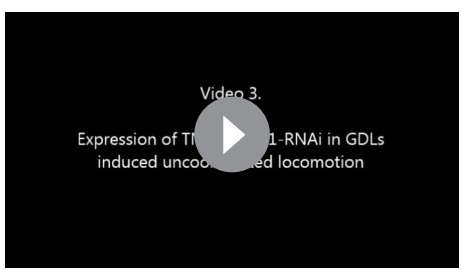

**Video 3.** Slow and uncoordinated locomotion in the third instar larvae expressing *TNT* or *Gad1-RNAi* in GDLs (*GDL-GAL4>UAS-TNT, GDL-GAL4>Gad1-RNAi*). (Related to *Figure 3*).

## A27h is an excitatory interneuron that drives motor neurons

The A27h neuron, which is the strongest GDL synaptic partner, arborizes in the motor domain, potentially driving motor neurons (*Figure 5—figure supplement 1B,C*). To determine which motor neurons A27h connects, we reconstructed the postsynaptic partners of A27h in an EM volume of the whole CNS. We found that A27h synapses bilaterally onto two identified motor neurons, aCC and RP5 (*Figure 5—figure supplement 2A*), which innervate longitudinal muscles via the intersegmental nerve (ISN), and also additional ISN motor neurons. We validated these findings by reconstructing these neurons

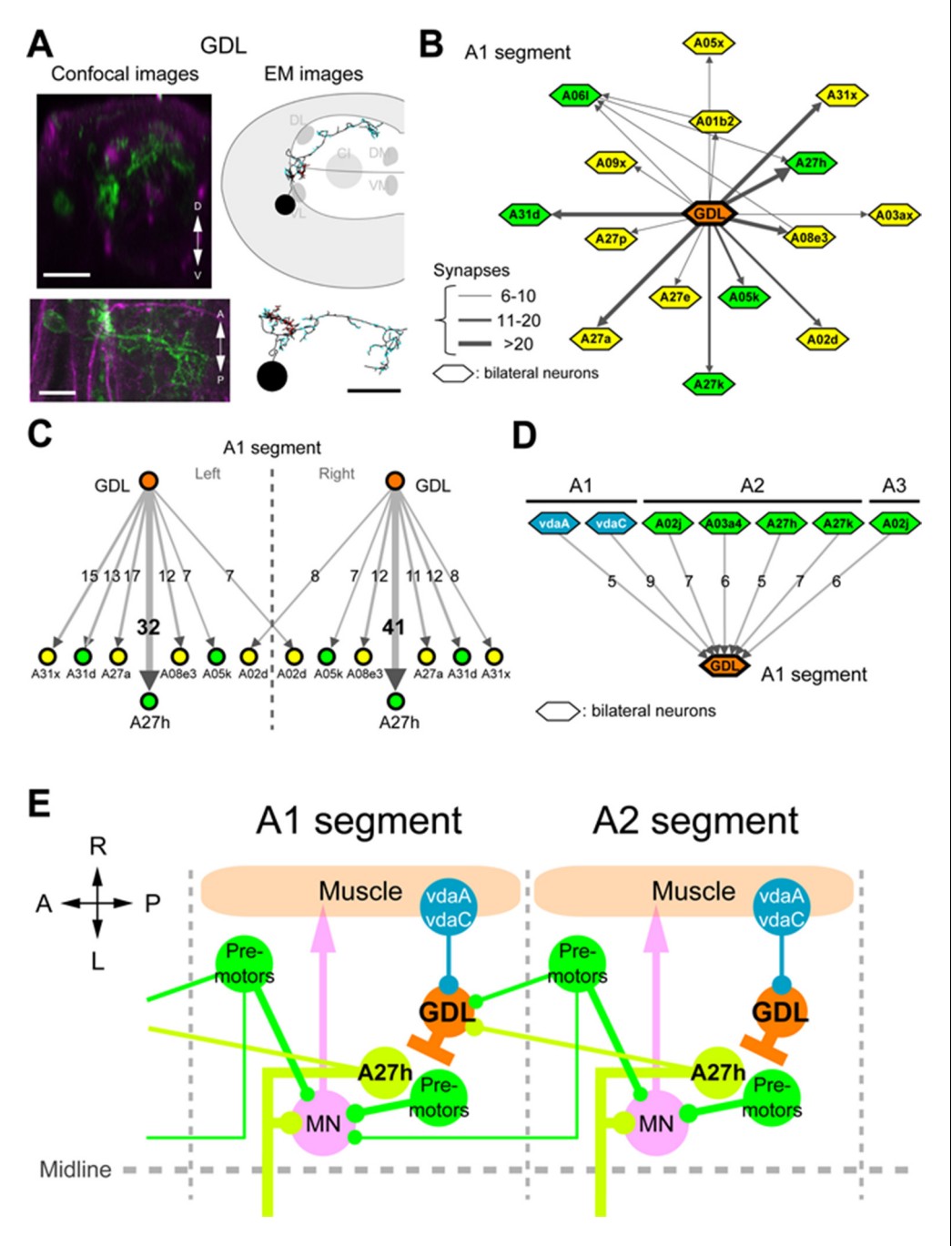

**Figure 4.** Circuit diagram around GDLs. (**A**) Comparing the confocal images (left) and EM reconstruction (right) of a GDL (top: cross-sectional view, bottom: dorsal view). Postsynaptic sites (cyan) and presynaptic sites (red) are shown in the EM images (right). Scale bar represents 20 μm (upper left), 10 μm (bottom). (**B**) The EM circuit graph of GDLs and their postsynaptic neurons. Hexagonal shape denotes a group of left-right homolog neurons. Connections with less than 6 synapses are not included (green: premotor neurons, yellow: others). (**C**) Major postsynaptic ("downstream") targets of a GDL in the abdominal segment A1. A27h is the strongest synaptic partner of a GDL. Numbers on the directed arrows indicate the number of synapse. (**D**) Major presynaptic ("upstream") targets of a GDL include two dendritic arborization (da) sensory neurons (blue). All other presynaptic targets identified are premotor neurons in the posterior segments (green). (**E**) A circuit model around GDLs. From the wiring diagram, a GDL has connections with several premotor neurons at both upstream and downstream. The symbols: NMJ (arrowheads), putative excitatory synapse (circles), and inhibitory synapse (bars). Thickness corresponds synaptic strength. (See also *Figure 4—figure supplement 1–3*.)

*Figure 4 continued on next page*

*Figure 4 continued*

The following figure supplements are available for figure 4:

**Figure supplement 1.** No GRASP signal was detected with motor neurons (Related to *Figure 4*).

**Figure supplement 2.** Morphology of the major presynaptic and postsynaptic neurons of GDL.

**Figure supplement 3.** Adjacency matrix for GDL circuits.

in an EM volume of a second larva (*Figure 5—figure supplement 2B*).

We tested whether A27h excites motor neurons by doing paired whole-cell recordings. In current clamp, we injected current into A27h to induce action potentials and recorded the membrane potential of an aCC motor neuron within the same segment. We found that the aCC motor neuron was efficiently depolarized in response to action potential generation in A27h (*Figure 5A–D*). The depolarizing response was with a very short delay ($\leq 5$ ms), consistent with the direct synaptic connection shown by the EM reconstruction (*Figure 5—figure supplement 2*). The efficiency with which A27h is capable of driving aCC correlates with the position of A27h presynaptic terminals, near the proximal portion of the aCC axon (*Figure 5—figure supplement 2C*), which is the presumptive spike initiation zone (*Gunay et al., 2015*). We also recorded the intrinsic activity of A27h and aCC and found that they were synchronized (*Figure 5E,F*). In order to determine the neurotransmitter used A27h, we first asked whether the expression pattern of *R36G02-GAL4* includes A27h neurons by driving the expression of myr-GFP (*Figure 5G*). We then used a photoactivatable GFP (*Ruta et al., 2010*) and identified the A27h axon within the *R36G02-GAL4* expression pattern by comparing it to the EM reconstructions (*Figure 5H*). Then, we labeled the presynaptic sites by driving synaptotagmin-HA and confirmed they were cholinergic using anti-ChAT antibody staining (*Figure 5I,J*). Acetylcholine is known to excite motor neurons in *Drosophila* larva (*Baines and Bate, 1998*; *Rohrbough and Broadie, 2002*).

Taken together, these results suggest that the neuron A27h induces muscle contraction. To test this, we targeted the expression of ChR2(T159C) to A27h using *R36G02-GAL4* and applied localized light to two segments in dissected larvae while monitoring muscle contractions along the body wall (see Materials and methods). We found that upon localized stimulation, muscles in the corresponding body wall segments contracted (*Video 4*). Although involvement of other neurons included in the *R36G02-GAL4* expression pattern cannot formally be excluded as being involved in this light activated muscle contraction response, these results provide strong support for the notion that A27h activates motor neurons and induces muscle contraction.

## A27h is active only in forward peristalsis

The sequential intersegmental connections between the inhibitory GDL and the excitatory A27h neurons (*Figure 4E*) suggest that A27h may be active synchronously with the peristaltic wave of motor neuron activity that propagates locomotion. To test this hypothesis, we monitored the activity of A27h neurons and aCC motor neurons during fictive locomotion (*Figure 6*). We observed a wavelike activity that propagates from posterior to the anterior segments (*Figure 6A*). Interestingly, unlike GDLs that are active during both forward and backward locomotion (*Video 5*), A27h was activated only during forward locomotion (*Figure 6B*). This suggests that though GDL participates in both forward and backward locomotion, the excitatory neuron A27h is specialized in forward locomotion. We postulate a different premotor neuron acts during backward locomotion and we found a possible candidate for which a genetic driver line does not exist (*Figure 8—figure supplement 1A* and see Discussion).

## GDLs are necessary for forward peristalsis and sufficient to interrupt it

The segmentally linked connections between inhibitory GDL neurons and excitatory A27h neurons in the next anterior segment (*Figure 4E*) suggest a mechanistic explanation for wave propagation in peristaltic locomotion. We hypothesize that the commands to contract one segment also promote the relaxation in the next anterior segment, and that contraction termination is coupled with circuit

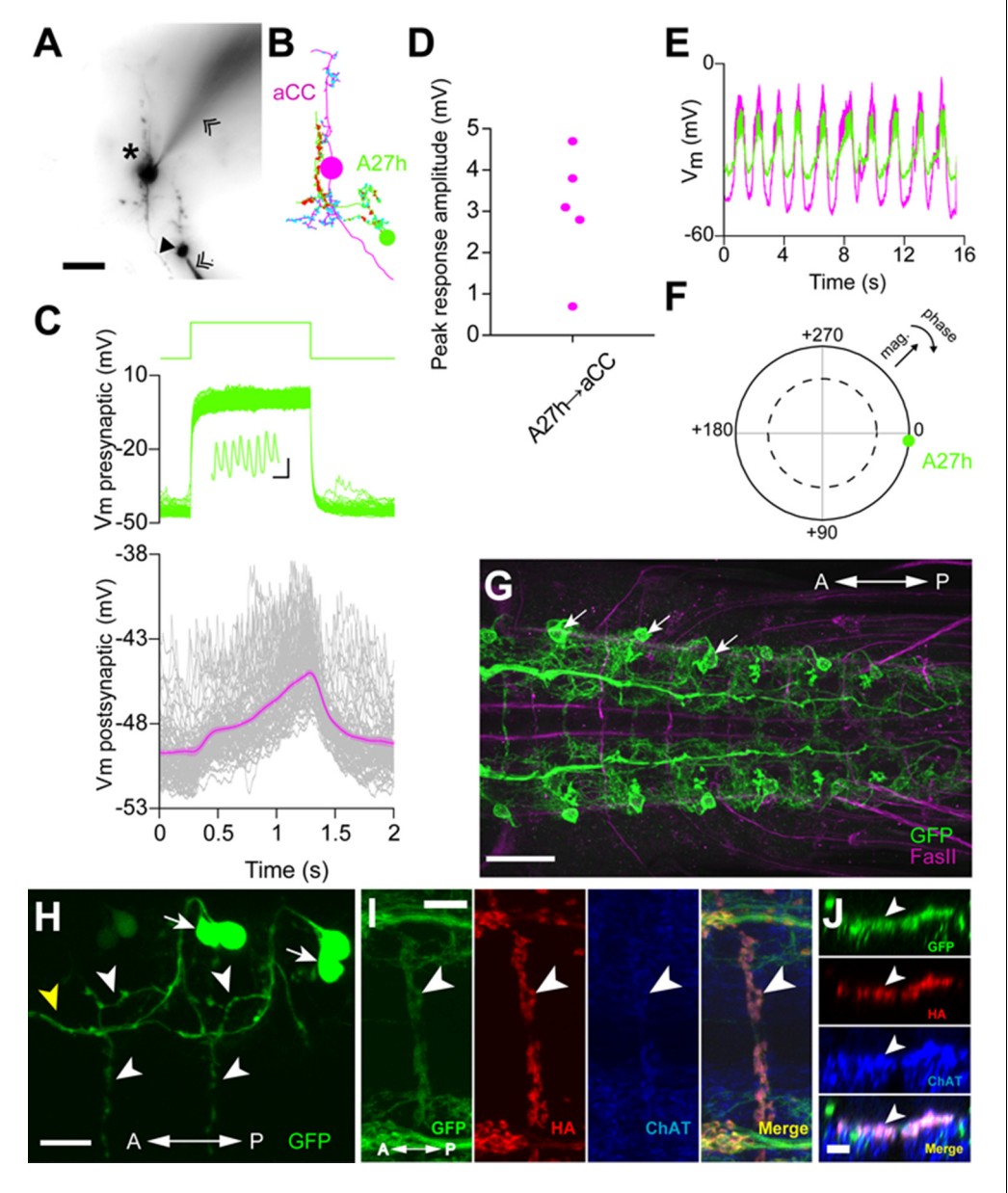

**Figure 5.** A27h is an excitatory premotor interneuron. (A) An example of a paired recording of an aCC motor neuron (asterisk) and a presynaptic A27h (arrowhead) dye-filled with Alexa 568 in the intracellular recording solution. Recording electrodes are indicated with chevrons. (B) EM reconstructions of aCC (magenta) and A27h (green). Input synapses are labeled in cyan, output synapses in red. (C) A current command (50 pA) results in A27h firing action potentials (see zoomed-in view in inset, scale bar indicates 10 ms, 0.5 mV), which efficiently drives the postsynaptic aCC motor neuron (magenta trace depicts mean of 100 trials ± SEM). (D) The maximum voltage response in aCC to presynaptic stimulation. Each point indicates the mean response of 100 trials of current injection in a different cell. (E) Endogenous activity patterns of these two cells, with each burst corresponding to a peristaltic wave. (F) Phase plot describing the coherency between the two cells, with magnitude of coherence depicted as the distance from the center, and the phase shift as deviation from 0° (with aCC at 0°). Dashed line indicates α = 0.05 for coherence magnitude statistically deviating from 0. (G) Expression driven by *R36G02-GAL4*. Assessed with the *10xUAS-IVS-myr::GFP* reporter and immunostaining with anti-GFP (green) and anti-FasII (magenta) antibodies. Strong expression was seen in A27h (arrows) and a small number of other cells in the VNC. (H) Photolabeling of A27h neurons. A flash of near-UV light (~405 nm) was applied to a dorso-lateral region of the VNC dissected from a *R36G02-GAL4>UAS-C3PA* larva, to label A27h and neighboring cells and their axonal arborization. The cell body of A27h can be uniquely identified for its stereotypic relative position to other cells

*Figure 5 continued on next page*

*Figure 5 continued*

(arrows); white arrowheads, axons of A27h; yellow arrowhead, an axon of a different cell. (I, J) A27h presynaptic terminals (arrowheads) express ChAT. Triple labeling for membrane-GFP (green), presynaptic marker (red) and ChAT (blue) (in *R36G02-GAL4>UAS-syt::HA;10xUAS-IVS-myr::GFP*). Dorsal (I) and cross-sectional (J) view are shown. Scale bar represents 30 μm in (G), 20 μm in (A, H), 10 μm in (I) and 5 μm in (J). (See also *Figure 5—figure supplement 1*, *2*).

The following figure supplements are available for figure 5:

**Figure supplement 1.** The connectivity of premotor neurons (Related to *Figure 5*).

**Figure supplement 2.** Bilateral A27h connection to motor neurons was confirmed in two independent EM volumes.

activity that enables contraction of the next segment. To test this hypothesis, we monitored the peristaltic waves following GDL activity perturbation during larval locomotion.

First, we observed that coordinated GDL activity is necessary for locomotion. We activated GDLs in all segments simultaneously by driving ChR2(T159C) (*Berndt et al., 2011*) with *GDL-GAL4*. All individual larvae stopped moving upon presentation of blue light (10 out of 10; [*Figure 7A* and *Video 6*]). Larval abdominal segments were paralyzed but, interestingly, they could still move their thoracic segments, which do not participate in peristaltic wave propagation. To control for a potential startle response to blue light (*Xiang et al., 2010*), we confirmed these findings using thermogenetics and dTRPA1 (*Pulver et al., 2009*). Larvae showed very slow and uncoordinated locomotion at a restrictive temperature at which dTRAPA1 expression is driven (32°C; p<0.001; *Figure 7B–D*). To determine the nature of this locomotion blockage, we activated all GDLs by ChR2(T159C) in a semi-intact preparation where we could monitor muscle contractions using mhc::GFP (*Hughes and Thomas, 2007*). We found that muscles relaxed when all GDLs were active (*Figure 7E*), contrary to the whole-body contraction (hunch) normally observed as part of the startle response elicited by blue light (*Ohyama et al., 2013*; *Vogelstein et al., 2014*). To exclude that neurons in the *GDL-GAL4* expression pattern other than GDLs played a role in this muscle relaxation, we used *tsh-GAL80* to suppress expression in abdominal segments, and this rescued the immobilization phenotype (*Video 6*). These results were confirmed using optogenetic CsChrimson-mediated activation of GDLs and a different driver line, *R15C11-LexA*; this resulted in similar phenotypes (*Figure 7—figure supplement 1* and *Video 6*).

Then, we determined that the suppression of GDL activity is indeed necessary for the propagation of the peristaltic wave. In a semi-intact preparation, we restricted blue light illumination to a window comprising two to three consecutive abdominal segments to excite GDLs for a few seconds using ChR2(T159C). This localized stimulation induced muscles relaxation in the corresponding body-wall segments and the disappearance of peristaltic waves (72%, 18/25 trials) only when the segments were illuminated at the front of the muscle contraction wave (*Figure 7F* and *Video 7*). Furthermore, upon removal of light, the wave sometimes resumed at the illuminated segments (16%, 4/25 trials) (*Video 7*). Illuminating segments more anterior to the front of the wave did not prevent the wave from propagating across them, but the wave appeared slower (12%, 3/25 trials) (*Video 7*). These results show that local GDL activation in a few segments at the front of the wave is sufficient to arrest the peristaltic wave.

Taken together, our results support a model of peristaltic wave propagation consisting of co-activation (e.g. A27h) of the motor neurons in one segment with the inhibitory neurons (e.g. GDL) that suppress activity of the homologous

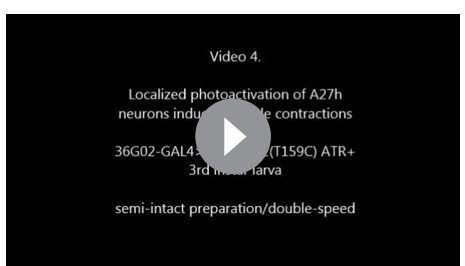

**Video 4.** Localized photoactivation of A27h neurons induced muscle contractions. ChR2-T159C was expressed in A27h neurons (*36G02-GAL4>UAS-ChR2-T159C*). A semi-intact larva preparation from third instar larva. Double-speed. (Related to *Figure 5*)

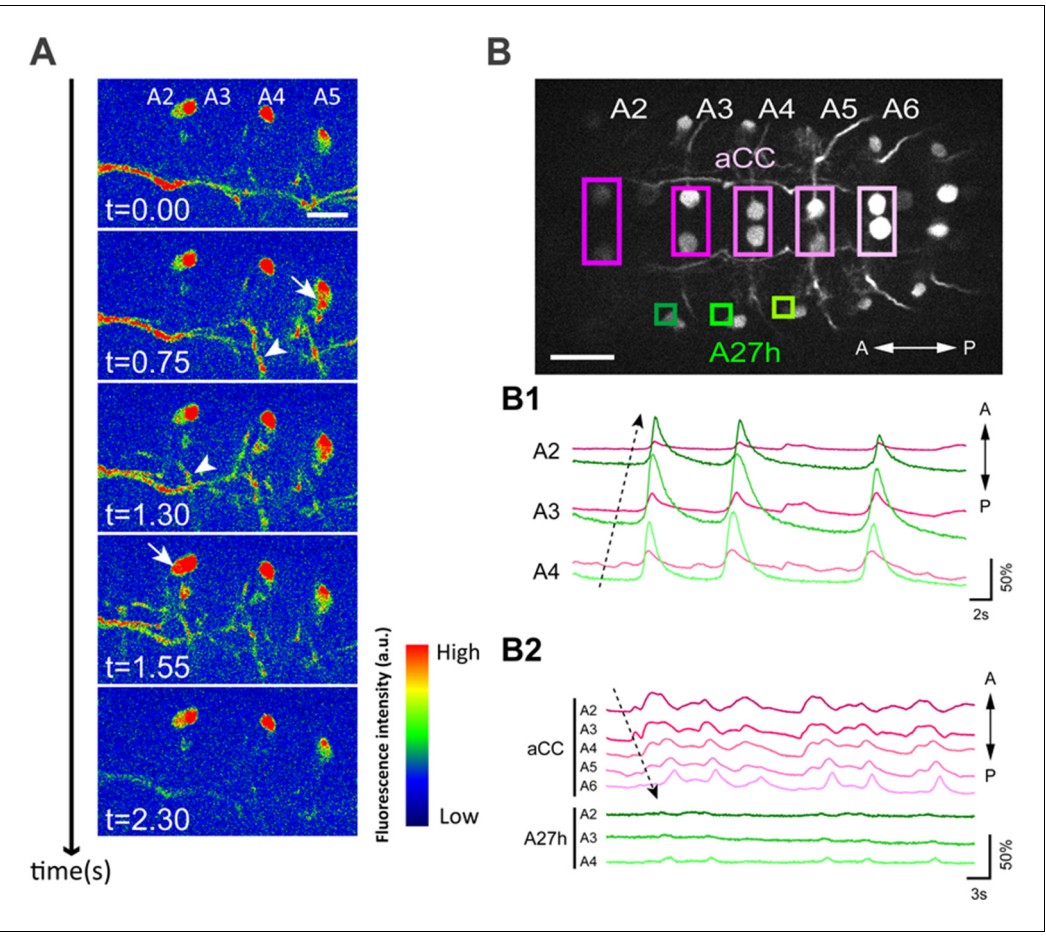

**Figure 6.** A27h participates in forward motor activity. (**A**) Calcium imaging of A27h (in *R36G02-GAL4>20xUAS-IVS-GCaMP6m*). Arrows denote the cell bodies of A27h neurons and arrowheads axons of A27h neurons. (**B**) Simultaneous imaging of the activity of A27h neurons (green) and aCC motor neurons (magenta) (in *R36G02-GAL4,eve-GAL4>20xUAS-IVS-GCaMP6m*). The top panel shows the region of interests (ROI) used for the analyses. (**B1, 2**) Dashed arrows denote the directions of motor activity. A27h was activated only during forward movement (**B1**) but not backward movement (**B2**). Scale bar represents 30 μm in (**B**) and 15 μm in (**A**).

**Video 5.** Simultaneous calcium imaging of A27h neurons and aCC motor neurons. GCaMP6m was expressed in A27h neurons and subsets of motor neurons (*36G02-GAL4, eve-GAL4>20xUAS-GCaMP6m*). A27h neurons are indicated by arrows. An isolated CNS preparation from third instar larva. Double-speed. (Related to *Figure 6*)

excitatory neurons (A27h) in the next segment (*Figure 8*).

## Discussion

We discovered a circuit whose structure and function provides a mechanism for understanding forward wave propagation in peristaltic locomotion. This circuit consists of a chain of alternating excitatory and inhibitory neurons spanning all abdominal segments. The core elements of the chain include just one excitatory and one inhibitory neuron per hemisegment. We demonstrate here that the inhibitory neuron (GDL) is sufficient to halt the peristalsis and to relax muscles in all segments, suggesting it is a point of coordination between forward and backward locomotion. We further demonstrate

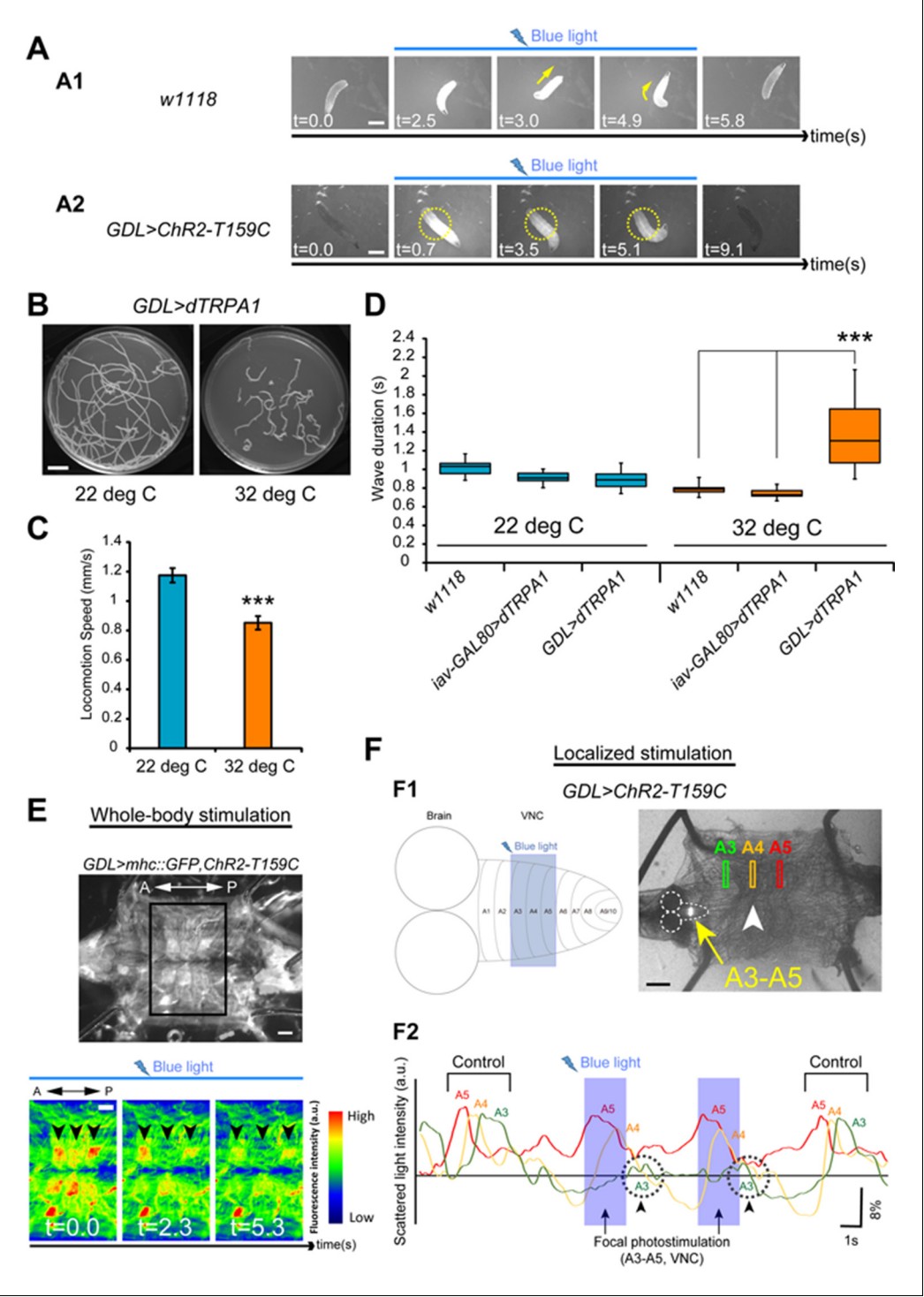

**Figure 7.** Optical perturbation of the activity of GDLs disrupts the peristalsis. (**A**) Behavioral responses induced by optogenetic activation of *GDL-GAL4*-expressing neurons (A1) A wild-type larva. Illumination with blue light (~480 nm) induced light-avoidance behaviors such as backward movement and head turning. (A2) Channelrhodopsin-2 (ChR2)-mediated activation of GDLs completely immobilized the abdominal segments of the larva (yellow dashed circles; 10 of 10 larvae [*GDL-GAL4>UAS-ChR2-T159C*] compared to 0 of 8 cases in the control larvae [*w1118>UAS-ChR2-T159C*]). (**B–D**) Larvae expressing dTRPA1 in *GDL-GAL4* showed locomotion defects at a restrictive temperature. Traces (**B**), locomotion speed (**C**) and wave duration (**D**) at permissive and restrictive temperatures are shown (C; Locomotion speed, 0.85 ± 0.05 mm/sec compared to 1.17 ± 0.05 mm/sec in the control larvae, the larvae with the same genotype at a permissive temperature (22°C), D; Wave duration, 1.42 ±

*Figure 7 continued on next page*

*Figure 7 continued*

0.43 sec [*GDL-GAL4>UAS-dTRPA1*] compared to 0.79 ± 0.06 sec [*w^{1118}*] and 0.74 ± 0.05 sec [*iav-GAL80>UAS-dTRPA1*]; p<0.001. Note that larvae normally crawl faster at 32°C than at 22°C). For all conditions in each figure, n = 20 in (C) and n = 10 in (D). (E) A dissected larva expressing ChR2-T159C in *GDL-GAL4* and mhc::GFP in muscles (*GDL-GAL4>mhc::GFP, UAS-ChR2-T159C*). When blue light was applied during peristalsis, contracted muscles became relaxed (n = 12). (F) (F1) Localized photostimulation was applied to an anterior portion of the VNC (around A3-A5, yellow arrow) during peristalsis. Arrowhead denotes the contracting segments at this moment. (F2) Muscular movement was examined by using the scattered light changes. The light intensity change in muscles in A3-A5 is plotted. In this example, the peristaltic wave halted at A3 (dashed circle with arrowheads). Statistical significance was determined by Student's *t*-test or one-way ANOVA followed by Tukey's test for multiple comparisons (\*\*\*p<0.001). Scale bar represents 15 mm in (C), 9 mm in (A), 250 μm in (F) and 200 μm in (B). (See also *Figure 7—figure supplement 1*.)

The following figure supplement is available for figure 7:

**Figure supplement 1.** Confirmation of the expression of ChR2 in GDLs.

---

that the excitatory neuron (A27h) is active during forward but not backward peristalsis, suggesting the existence of another excitatory circuit component critical for backward peristalsis among the synaptic partners of the GDL inhibitory neuron. This circuit defines a backbone of repeating, connected, modules for excitation and inhibition similar to those postulated in a computational model for peristalsis (*Gjorgjieva et al., 2013*) on the basis of behavioral observations that predicted the existence of central pattern generators (*Suster and Bate, 2002*).

We found that the excitatory neuron (A27h) is premotor, directly synapsing onto motor neurons of its own segment only and that control both dorsal and ventral longitudinal muscles. This suggests an explanation for the observation that in forward crawling, dorsal and ventral longitudinal muscles contract simultaneously (*Heckscher et al., 2012*). In backward peristalsis, however, a phase gap has been observed in the timing of dorsal and ventral muscle contraction (*Heckscher et al., 2012*). This decoupling could require a more complex circuit structure for backward wave propagation, and therefore suggests an explanation for the lack of an equivalent excitatory neuron in the circuit chain for backward peristalsis. We found, however, neurons postsynaptic to the inhibitory neuron (GDL) whose anatomy and position in the circuit suggest a role in backward peristalsis (*Figure 8—figure supplement 1A*). In contrast, the inhibitory neuron (GDL) itself does not synapse onto motor neurons, and therefore occupies a higher-order position in the circuit that allows its participation in both forward and backward wave propagation in peristalsis. Furthermore, the GDL axon targets the intermediate lateral neuropil, which is neither

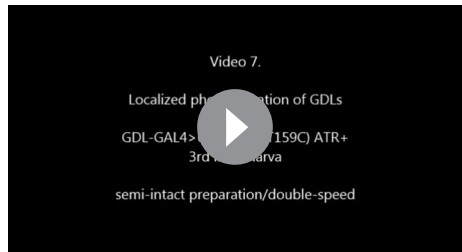

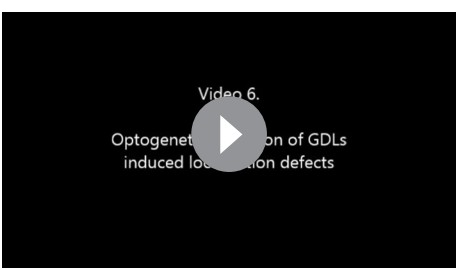

**Video 6.** Optogenetic activation of GDLs induced locomotion defects. Behavior of first or third instar larvae expressing ChR2-T159C (*GDL-GAL4>UAS-ChR2-T159C*) or CsChrimson (*R15C11-LexA>LexAop2-CsChrimson*) in GDLs, upon light application. (Related to *Figure 8*)

**Video 7.** Localized activation of GDLs affected larval peristalsis. ChR2-T159C was expressed in GDLs (*GDL-GAL4>UAS-ChR2-T159C*). (I) Localized photoactivation of GDLs in a portion of VNC during peristalsis halted the peristaltic wave at the corresponding region in the body wall. (II) The wave sometimes resumed at the illuminated segments. (III) Illuminating segments more anterior to the front of the wave did not prevent the wave from propagating. A semi-intact larva preparation from third instar larva. Double-speed. (Related to *Figure 8*)

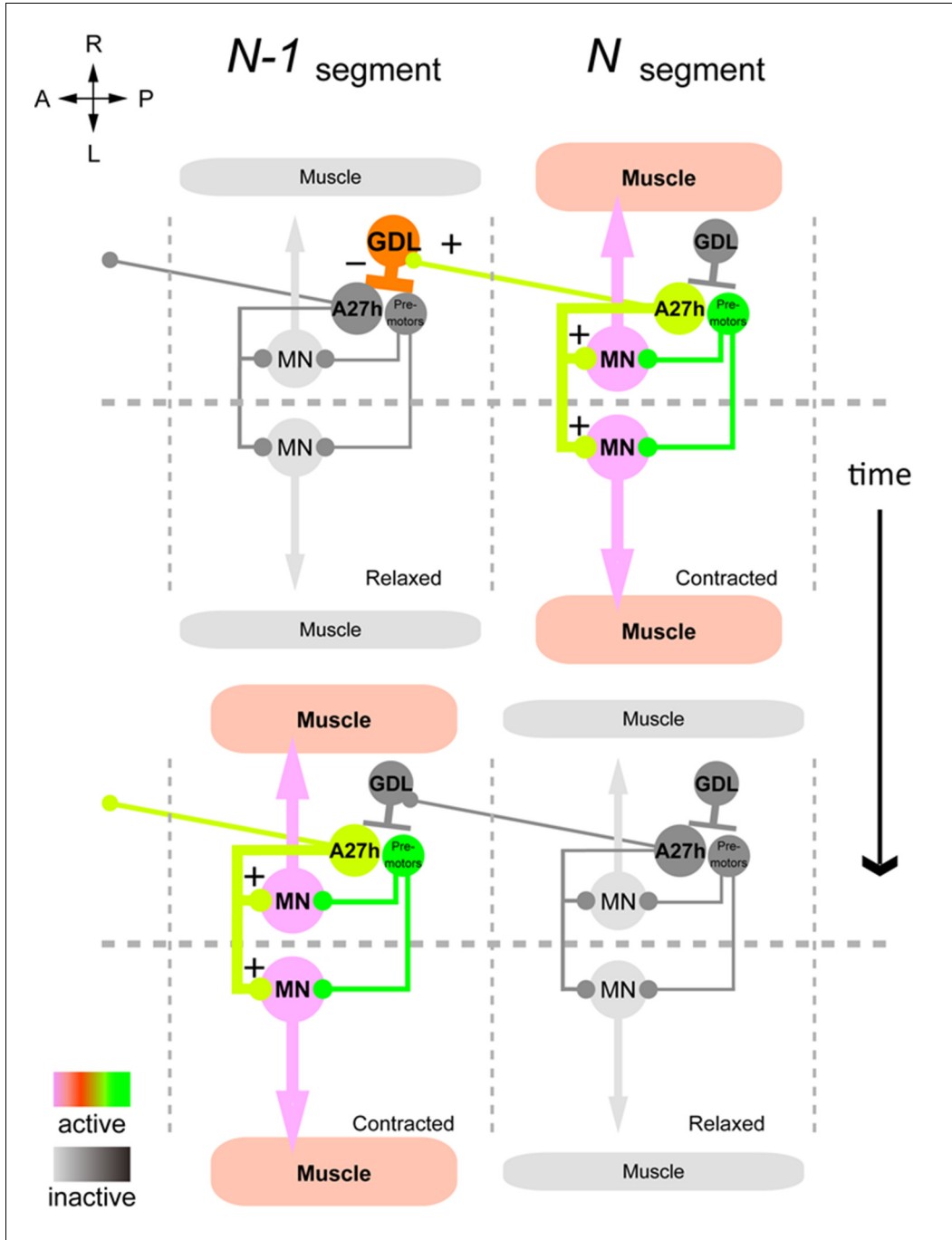

**Figure 8.** Summary of the GDL circuit. The information flow in the GDL-A27h premotor circuit. At a time point during forward peristalsis when A27h in segment *N* is active and driving motor activity in the segment, GDL in the next anterior segment *N-1* is active and inhibits the activity of A27h and the downstream motor activity in the segment. As the motor wave propagates anteriorly and motor activity in segment *N* declines, so does the GDL in segment *N-1*, thus releasing the target A27h from its inhibition (gray: inactive, other colors: active). (See also *Figure 8—figure supplement 1*.)

The following figure supplements are available for figure 8:

**Figure supplement 1.** A proposed circuit mechanism for moderating peristaltic locomotion.

**Figure supplement 2.** Synaptic relations of GDL and A27h with known larval interneurons.

in the domain of motor neuron dendrites nor in the somatosensory domain, suggestive of a role higher-order motor coordination. Relevant for forward peristalsis, GDL disinhibits the excitation of its anterior homologs, by removing inhibition from a glutamatergic interneurons (A02j) implicated in the regulation of peristaltic speed (one of the PMSIs; [*Kohsaka et al., 2014*]). A02j is presynaptic to GDLs in anterior segments (*Figure 4D* and *Figure 8—figure supplement 2A*).

A model of peristaltic locomotion must consider the coordination of left and right hemisegments (*Gjorgjieva et al., 2013*). Though we found that the chain of alternating inhibitory and excitatory neurons runs independently on the left and right sides of the body, the excitatory neuron (A27h) presents a bilateral arbor and drives motor neurons bilaterally. Our wiring diagram best supports a model of left-right coordination where excitatory neurons communicate with each other (*Gjorgjieva et al., 2013*), but with the caveat that this synergy takes place by the simultaneous co-activation of the target motor neurons rather than reciprocal excitation. This model has been shown to support longer contraction episodes at the front of the wave (*Gjorgjieva et al., 2013*), consistent with observations of muscle contraction in peristalsis (*Heckscher et al., 2012*). Independently of the timing, the fine-tuning in the intensity of left-right contractions has been shown to be under control of Even-skipped+ evolutionarily conserved neurons, which integrate both proprioceptive inputs and motor commands (*Heckscher et al., 2015*).

The dissected larval CNS undergoes spontaneous waves of motor neuron activation at about 1/10[th] the normal speed (*Fox et al., 2006*; *Pulver et al., 2015*). These waves occur in the absence of sensory feedback, indicating the presence of CPGs and also suggesting a role for sensory feedback in speeding up the peristaltic wave (*Suster and Bate, 2002*). The circuit chain of excitatory and inhibitory neurons described here could be a part of the CPG, and we additionally found these neurons are modulated by proprioceptive inputs (from vpda class I dendritic arborization neuron; *Figure 8—figure supplement 1B*). Given that the vpda is a stretch receptor (*Cheng et al., 2010*; *Tamarkin and Levine, 1996*), it would be active in the segment ahead of the wave of contraction, which is being stretched by the pull exerted by the contracting segment (*Figure 8—figure supplement 1B*). Proprioceptive feedback action onto the excitatory neuron of the circuit chain could then have two simultaneous effects: promotion of the contraction in the segment ahead of the wave (via activation of A27h), and relaxation of the segment twice removed (via activation of GDL, which acts on the segment anterior to it; *Figure 8—figure supplement 1C*). We also found two somatosensory neurons (vdaA and vdaC) synapse axo-dendritically onto the premotor excitatory neuron (A27h) and axo-axonically onto the inhibitory neuron (GDL) in their own segment (*Figure 8—figure supplement 1B*). Although the function of these two sensory neurons remains unclear, we speculate that this axo-axonic, likely depolarizing, connection onto GDL reduces the membrane action potential of its axon, reducing synaptic release of GABA onto A27h in the same segment (*Burrows and Matheson, 1994*). Our model refines a previous model where the proprioceptive feedback was thought to signal the successful contraction of a segment (*Hughes and Thomas, 2007*). We suggest that, in addition, at least some of the proprioceptive feedback (vpda) facilitates wave propagation and, therefore, may underlie the reduction in speed observed in fictive crawling (*Fox et al., 2006*; *Pulver et al., 2009*).

In addition to the excitatory premotor interneuron A27h, we found two other interneurons that receive direct synaptic inputs from a GDL (A02d and A08e3) and that, like A27h, also integrate inputs from stretch receptors (vpda, dbd and vbd; *Figure 8—figure supplement 2B,C*). One interneuron (A08e3) is an Even-Skipped+ neuron that maintains left-right symmetric muscle contraction amplitude (*Heckscher et al., 2015*). The other (A02d) is a glutamatergic interneuron that belongs to a lineage of neurons thought to mediate speed of locomotion (one of the PMSIs; [*Kohsaka et al., 2014*]). While A02d is a segment-local interneuron, proprioceptive axons span multiple segments (*Merritt and Whitington, 1995*; *Schneider-Mizell et al., in press*), suggesting that a GDL can suppresses the effect of proprioceptive feedback specifically within its own segment without affecting the relay of proprioception to adjacent segments. Furthermore, A02d synapses onto a glutamatergic interneuron (A08a) thought to contribute to muscle relaxation in the wake of the peristaltic wave (*Itakura et al., 2015*), which could be mediated via putative GABAergic premotor neurons (A31d; *Figure 8—figure supplement 2B*). Taken together, we suggest that one of the functions of the inhibitory neuron GDL is to gate proprioceptive feedback within its segment which has implications for the control of both speed and posture (*Heckscher et al., 2015*).

Finally, we observed a descending neuron from the SEZ that synapses onto the excitatory neuron (A27h) of the circuit chain in all segments (*Figure 8—figure supplement 1D*). This motif has been observed and modeled in the leech and crayfish, where it enables the modulation of wave propagation speed (*Acevedo et al., 1994*; *Cacciatore et al., 2000*; *Smarandache et al., 2009*; *Stein, 1971*; *Wiersma and Ikeda, 1964*). The brain and SEZ have been deemed non-essential for wave propagation (*Berni et al., 2012*). Speed of wave propagation, therefore, may be controlled in at least two ways: by proprioceptive feedback and by descending inputs. The existence of a circuit chain formed by excitatory and inhibitory neurons might be all that remains when both sensory feedback and the brain are absent, explaining the existence of wave propagation in decerebrated animals (*Berni et al., 2012*), and even for a small set of isolated abdominal segments (*Pulver et al., 2015*).

## Materials and methods

### Fly strains

The following fly strains were used: $w^{1118}$ (Bloomington stock number: #6326) (*Hoskins et al., 2001*), 9-20-GAL4 (*Hughes and Thomas, 2007*), eve(RRa)-GAL4 (*Fujioka et al., 2003*), R36G02-GAL4 (#49939), OK6-LexA (*Kohsaka et al., 2014*), R15C11-LexA (#52492), UAS-mCD8::GFP (#5137) (*Lee and Luo, 1999*), 10xUAS-IVS-mCD8::GFP (#32185, #32186) (*Pfeiffer et al., 2010*), 10xUAS-IVS-myr::GFP (#32197, #32198) (*Pfeiffer et al., 2010*), 10xUAS-IVS-mCD8::RFP,13xLexAop2-mCD8::GFP (#32229) (*Liu et al., 2012*), UAS-CD4::spGFP1-10 (*Gordon and Scott, 2009*), LexAop-CD4::spGFP11 (*Gordon and Scott, 2009*), UAS-syt::GFP (*Zhang et al., 2002*), UAS-syt::HA (*Robinson et al., 2002*), 20xUAS-IVS-GCaMP6m (#42748, #42750) (*Chen et al., 2013*), UAS-dTRPA1 (*Pulver et al., 2009*), UAS-TNT (#28838) (*Sweeney et al., 1995*), UAS-IMPTNT(V1) (#28840) (*Sweeney et al., 1995*), 13xLexAop2-IVS-CsChrimson-mVenus (#55139), UAS-C3PA-GFP (*Ruta et al., 2010*), mhc::GFP/Cyo (*Hughes and Thomas, 2007*), tsh-GAL80/Cyo (*Clyne and Miesenbock, 2008*), Gad1-RNAi (#28079(VALIUM10), #51794(VALIUM20)) and Dicer-2 (#24650, #24651). Flies were raised on conventional cornmeal agar medium at 25°C except the following: in order to enhance RNAi potency, the transgenic fly (UAS-Gad1-RNAi (VALIUM10)) was combined with Dicer-2 and reared at a higher temperature (29°C).

### Transgenic flies

To generate *iav (inactive)-GAL80* transgenic line, we excised the GAL80 sequence from pBPGAL80Uw-6 (*Pfeiffer et al., 2010*) using BamHI and XbaI and subcloned the DNA between BamHI and StuI (blunt-ended) sites of *iav-GAL4* (*Kwon et al., 2010*). The resulting construct was used to transform $w^{1118}$ embryos using standard *Drosophila* micro-injection techniques (BestGene Inc). To generate *UAS-ChR2(T159C)* transgenic line, we first introduced KpnI and AgeI sites between the SwaI (12079) and PmeI(12095) sites of pJFRC2-INS (Plasmid #26215). We excised the sequence between the HindIII(6488) and XbaI(8490) sites of pJFRC2-INS (Plasmid #26215) and replaced with the sites of *pJFRC7-20XUAS-IVS-mCD8::GFP* (Plasmid #26220, XbaI(8740) and HindIII(6488), 4*5xGAL4_DBD). Then, we replaced the sites between XhoI(7341) and XbaI(8740) with *Drosophila* codon-optimized ChR2(T159C)::YFP synthesized by Biobasic inc. We next excised the sequence between the HindIII and PacI sites of the plasmid and amplified by PCR using primers 5-AgeI (CATGCGCACCGG TGGCCAGGGCCGCAAG) and 3-KpnI (CACTTGGTACCTGGCCATTAATTAAGGCCGGCC). The resulting construct was used to transform *y[1] w[67c23]; P(CaryP)* attP40 or attP2 sites as described above.

### Immunocytochemistry

Dissected larvae were fixed in phosphate buffered saline (PBS, NaCl 137 mM, KCl 2.7 mM, $Na_2HPO_4$ 8.1 mM, $KH_2PO_4$ 1.5 mM, pH7.3) containing 4% paraformaldehyde for 30 min at room temperature. After two 15 min washes with 0.2% Triton X-100 in PBS (PBT), the larvae were incubated with 5% normal goat serum in PBT for 30 min. The larvae were then incubated overnight at 4°C with the primary antibody. After two 15 min washes, the larvae were incubated overnight at 4°C with the secondary antibody. Images were acquired using a confocal microscope (FV1000, Olympus, Japan). Primary antibodies were used at the following dilutions: rabbit anti-GFP (cat# Af2020, Frontier Institute; 1:1000), mouse anti-GFP (cat# G6539, Sigma; 1:100), guinea pig anti-GFP (cat# Af1180,

Frontier Institute; 1:1000), rabbit anti-HA (cat# C29F4, Cell Signaling Technology; 1:1000), rabbit anti-DsRed (cat# 632496, Clontech; 1:1000), mouse anti-FasII (mAB-1D4, Hybridoma Bank, University of Iowa; 1:10), rabbit anti-GABA (A2052, Sigma; 1:100), mouse anti-ChAT (mAB-4B1, Hybridoma Bank, University of Iowa; 1:50). Secondary antibodies were used at the following dilutions: Alexa Fluor 488 or Cy3-conjugated goat anti-rabbit IgG (A-11034 or A-10520, Invitrogen Molecular Probes; 1:300), Alexa Fluor 555 or Cy5-conjugated goat anti-mouse IgG (A-21424 or A-10524, Invitrogen Molecular Probes; 1:300), and Alexa Fluor 488-conjugated goat anti-guinea pig IgG (A-11073, Invitrogen Molecular Probes; 1:300).

## Behavioral analysis

We conducted two locomotion assays. One is automated tracking of the trajectory of larval behavior and the other is manually measuring the duration of each peristaltic wave. For automated tracking, wandering third instar larvae were picked up and then transferred to an agar plate (90 mm in diameter) for acclimation (3 min). The larvae were then videotaped using a digital camera (GE60, Library, Japan) and tracked using the open-source ImageJ plugin wrMTrck (http://www.phage.dk/plugins/wrmtrck.html). Each video containing 20 larvae was recorded five times at 30 frames/sec for 3 min. The average speed of larval locomotion was calculated by dividing the total path length of the larvae by time. For manual analysis, wandering third instar larvae were gently washed in deionized water and then placed on an agar plate. After acclimation (3 min), the movements of the larvae were videotaped under a microscope (SZX16, Olympus, Japan) using an XCD-V60 CCD camera (30 frames/sec for 30 s) and the movies were downloaded into VFS-42 (Vision Freezer, Chori imaging). The wave duration, which is elapsed time between the landing of the posterior end and elongation of the head, was manually measured in the movies using Fiji (10 waves per larva). The frequency of larval locomotion (number of forward waves) was also manually calculated by dividing the total number of forward waves of each larva by the total time.

## Calcium imaging

Two types of microscopy were used for the measurement of neural activity, one for low magnification and the other for high-magnification imaging. Low-magnification imaging was performed on semi-intact preparation of wandering third instar larvae, in order to observe both the propagation of muscular contraction and calcium signals in the CNS. The larvae were pinned on a sylgard-coated dish (Silpot 184, Dow Corning Toray) and dissected in an external saline (NaCl 135 mM, KCl 5 mM, $MgCl_2 \cdot 6H_2O$ 4 mM, $CaCl_2 \cdot 2H_2O$ 2 mM, TES 5 mM, Sucrose 36 mM (pH7.1)) (*Marley and Baines, 2011*). The internal organs were removed without scratching the ventral nerve cord (VNC) and axons. To fix the position of the VNC, a pin was placed between the brain and the mouth hook. Imaging was performed on a fluorescence microscope (MVX10, Olympus, Japan) equipped with a CCD camera (XCD-V60, Sony, Japan) and 1x~4x objective lens. The images were acquired and downloaded into VFS-42 (Vision Freezer, Chori imaging) at 30 frames/sec, 640 x 480 pixels. High magnification imaging was performed on isolated CNS preparation. The third instar larvae were dissected in the external saline described above and the peripheral nerves were cut carefully to isolate the CNS. The isolated CNS was adhered to a double-sided tape (NW-K15, Nichiban, Japan) on a clean glass slide in the saline. Imaging was performed on an upright microscope (Axioskop2 FS, Zeiss, Germany) equipped with a spinning disk confocal unit (CSU21, Yokogawa, Japan), an EMCCD camera (iXon, Andor Technology, Germany) and a 40x or a 63x water objective lens. The images were acquired at 20 frames/sec. Fiji was used for image analyses and pseudocolored images.

## Optogenetic experiments

Parental flies were reared in an egg collection cup with an agar plate with yeast paste at 25°C. Eggs were laid for 1 hr and transferred to another agar plate with yeast paste containing 1 mM all-trans retinal (R2500, Sigma). The larvae were picked up and gently washed in deionized water. Then, they were placed on an apple agar plate and stimulated with blue light (for ChR2(T159C); band-pass filtered at 460–490 nm, ~400 µW/mm$^2$) or yellow light (for CsChrimson; band-pass filtered at 540–580 nm, ~1 mW/mm$^2$) using a conventional Hg arc lamp under a fluorescence microscope (SZX16, Olympus, Japan). The larvae were videotaped before and after stimulation using an XCD-V60 CCD camera (30 frames/sec for 1 min). Localized photostimulation was performed as described previously

(*Matsunaga et al., 2013*). Briefly, the VNC was exposed from the larvae (without scratching the axons as described above) and Argon laser (488 nm) was applied to a few segments of the VNC under a confocal microscope (FV1000, Olympus, Japan). The movement of the dissected larva was videotaped using a XCD-V60 CCD camera (30 frames/sec for 5 min).

## Temperature shift experiments

Third instar larvae were picked up and gently washed in deionized water. For the conditional activation assay using dTRPA1, the larvae were transferred from an agar plate at the permissive temperature (PT, 22°C) to a new agar plate at a restrictive temperature (RT, 32°C) on a heat plate (Thermo Plate, Tokai Hit, Japan). The larvae were videotaped at PT or RT conditions using an XCD-V60 CCD camera (30 frames/sec for 1 min).

## Photolabelling neurons using PA-GFP

To label the neurons expressing photoactivatable green fluorescent protein (PA-GFP), we used a conventional confocal microscope (FV1000, Olympus, Japan) equipped with 63x water objective lens and 405 nm violet (near-UV) laser. In order to fix the sample, we used an isolated CNS preparation, which was adhered to a double-sided tape on clean glass slide with the saline. We then defined the region of interest (ROI: the size 100x100 pixels) and stimulated 10 s. After 5 min (for stable photoactivation), cells were imaged with the same confocal microscope under 488 nm excitation.

## Electrophysiology

Larvae were dissected and central neurons accessed as described previously (*Baines and Bate, 1998*). Briefly, the larval CNS was removed and pinned onto a sylgard-coated dish using fine wire ("0.001 Tungsten 99.95% wire", California Fine Wire Company). A small section of the glial sheath surrounding the VNC between segments A2-A4 was ruptured using protease (0.1–1% Protease XIV, Sigma-Aldrich) dissolved in external saline (the same as above [*Marley and Baines, 2011*]), to expose cell bodies underneath. The preparation was viewed with a 60x/1NA water-dipping objective on a microscope (BX51WI, Olympus, Japan). GFP-expression mediated by *R36G02-GAL4, 10xUAS-IVS-myr::GFP* was used to identify A27h, and bright-field microscopy to identify aCC, with post hoc confirmation of cell identity by filling with 100 μM Alexa Fluor 568 hydrazide (Invitrogen Molecular Probes), which was included in the internal saline (MgCl2 · 6H$_2$O 2 mM, EGTA 2 mM, KCl 5 mM, HEPES 20 mM, K-D-Gluconic acid 140 mM). Whole-cell recordings were performed using standard thick-walled borosilicate electrodes (GC100TF-10; Harvard), fire-polished to resistances of 8–12 MΩ. Recordings were made using an Axon Multiclamp 700B amplifier with two CV-7B headstages, and digitized using a Digidata 1550. Traces were recorded using pClamp 10 (all from Molecular Devices), digitized at 20 kHz and filtered at 2 kHz. Data were analyzed using Clampfit 10 (Molecular Devices) and Spike2 (Cambridge Electronic Design).

## Coherence analysis of periodic activity

To determine the phase relationship between periodic signals in paired whole-cell recording experiments, we used direct multi-taper estimates of power spectra and coherency, as described before (*Pulver et al., 2015*). Briefly, we determined the dominant frequency of activity in aCC by examining its power spectrum, and then estimated coherence between signals in aCC and A27h. All spectral calculations were carried out using custom scripts written in MATLAB, now freely available online (https://github.com/JaneliaSciComp/Groundswell).

## EM reconstruction using CATMAID

EM reconstruction was performed as described previously (*Ohyama et al., 2015*; *Schneider-Mizell et al., in press*) using a modified version of CATMAID (*Saalfeld et al., 2009*). We manually traced the axonal and dendritic processes of GDLs or A27h neurons and identified the location of the pre- and post-synapses. We then reconstructed the presynaptic and postsynaptic neurons from the synaptic sites.

## Finding identified neurons in the EM volume

A genetic driver line such as a GAL4 line drives expression in a specific subset of neurons. Expression patterns of interest are generally sufficiently sparse that individual neurons can be located relative to gross landmarks (see for example [*Li et al., 2014*]) such as the entry points of nerves or lineages into the neuropile, which are highly stereotyped (*Cardona et al., 2010*). Each lineage in the Drosophila larval nerve cord about 10 to 15 neurons, each with a distinctive arbor. In the EM, we locate the entry point into the neuropile of the lineage bundle and then swiftly reconstruct the low-order branches (the "backbone" containing continuous microtubule; [*Schneider-Mizell et al., in press*]). Then these partial reconstructions are compared to the light- microscopy images of GAL4 expression patterns, and by a process of elimination the neuron of interest is easily found. These identified neurons are then reconstructed in full, and the position of the presynaptic varicosities is compared to those observed in the light microscopy volumes, to further confirm their identification. Then, each identified neurons is used as a starting point to reconstruct all their presynaptic and postsynaptic partner neurons. These additional neurons are then readability available for comparisons with light microscopy volumes or with other segment in the nerve cord.

## Statistical analysis

We analyzed the data using Student's *t* test and one-way analysis of variance (ANOVA) followed by Tukey's tests for multiple comparisons. Statistical significance is denoted by asterisks: ***$p<0.001$; **$p<0.01$; *$p<0.05$; n.s., not significant. All statistical tests were performed using R-project software (http://www.r-project.org). The results are stated as mean ± s.d., unless otherwise noted.

## Acknowledgements

We are grateful to Drs. Craig Montell, Gerald Rubin, Gero Miesenbock, John Thomas, Karl Deisseroth, Kristin Scott, Leslie Griffith, Loren Looger, Miki Fujioka, Richard Axel and Bloomington and Kyoto Stock Center for the gifts of stocks and reagents. We would like to thank Dr. James Truman and Janelia Fly Light Project for the imagery of genetic driver lines, and Drs. Alex Kolodkin, Chris Q Doe, Marco Tripodi, Marta Zlatic and Matthias Landgraf for helpful comments. The transgenic lines were generated by BestGene Inc. We thank the Fly EM Project Team at HHMI Janelia for the gift of the EM volume, the HHMI visa office, and HHMI Janelia for funding.

## Additional information

### Funding

| Funder | Grant reference number | Author |
|---|---|---|
| Global COE Program: the Physical Sciences Frontier | | Akira Fushiki |
| JSPS Research Fellowships for Young Scientists | | Akira Fushiki |
| Howard Hughes Medical Institute | | Akira Fushiki<br>Maarten F Zwart<br>Richard D Fetter<br>Albert Cardona |
| MEXT/JSPS KAKENHI | 22115002 | Akinao Nose |
| MEXT/JSPS KAKENHI | 221S0003 | Akinao Nose |
| MEXT/JSPS KAKENHI | 15H04255 | Akinao Nose |

The funders had no role in study design, data collection and interpretation, or the decision to submit the work for publication.

### Author contributions

AF, Conception and design, Acquisition of data, Analysis and interpretation of data, Drafting or revising the article, Contributed unpublished essential data or reagents; MFZ, Acquisition of data, Analysis and interpretation of data; HK, RDF, Analysis and interpretation of data, Contributed

unpublished essential data or reagents; AC, Conception and design, Analysis and interpretation of data, Drafting or revising the article, Contributed unpublished essential data or reagents; AN, Conception and design, Drafting or revising the article

**Author ORCIDs**

Albert Cardona, http://orcid.org/0000-0003-4941-6536

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
