## [Decision Letter]

Thank you for submitting your work entitled "A circuit mechanism for the propagation of waves of muscle contraction in *Drosophila*" for consideration by *eLife*. Your article has been reviewed by three peer reviewers (Leslie Griffith, Marco Tripodi and Brian Mulloney), and the evaluation has been overseen by Leslie Griffith as Reviewing Editor and Eve Marder as the Senior Editor.

The reviewers have discussed the reviews with one another and the Reviewing Editor has drafted this decision to help you prepare a revised submission.

Summary:

This paper provides an outline of the first unified circuit diagram for intersegmental coordination of forward peristalsis in the *Drosophila* third instar larva. In spite of its outstanding genetic advantages, in the realm of locomotor circuits the fly had lagged seriously behind other model organisms in which electrophysiology allowed circuit mapping. For many years investigators had to satisfy themselves with access only to the motor neuron effectors. In the last couple years, there has been a wave of papers implicating a few specific interneurons in locomotion, but nothing that gave a big picture of how the circuit worked. The breakthrough here is the marriage of the EM level anatomy with physiology and imaging. This is a well-designed pioneering work that elucidates fine-mechanisms of wave propagation in *Drosophila* larvae. In addition, it paves the way to the genetic and anatomical dissection of the CPG on a scale never approached before.

Essential revisions:

The reviewers were unified in thinking that the data presented were important and largely complete. The major suggestion for improvement of the paper is to place this work more carefully into the constellation of knowledge about coordinating circuits in arthropods so readers will be able to appreciate what the authors have accomplished.

1) Introduction, second paragraph et seq.: "A complete wiring diagram with synaptic resolution…" – good, and true. This goal has recently been achieved, using different methods, and published in these papers:a) Smarandache CR, Hall WM, and Mulloney B (2009) Coordination of rhythmic motor activity by gradients of synaptic strength in a neural circuit that couples modular neural oscillators. J. Neurosci. 29:9351-9360.b) Smarandache-Wellmann CR, and Grätsch S (2014) Mechanisms of coordination in distributed neural circuits: Encoding coordinating information. J. Neurosci. 34:5627-5639.c) Smarandache-Wellmann CR, Weller C, and Mulloney B (2014) Mechanisms of coordination in distributed neural circuits: Decoding and integration of coordinating information. J. Neurosci. 34:793-803.

2) The significance of this coordinating circuit's structure for effective locomotion has been analyzed quantitatively in:

Zhang C, Guy RD, Mulloney B, Zhang Q, and Lewis TJ (2014) The neural mechanism of optimal limb coordination in crustacean swimming. PNAS 111:13840-13845.

3) Discussion, last paragraph: “descending neurons from the SEZ […] motif has been modelled in the leech, where…" – interesting observation, and a fair reference. More immediately relevant to work in *Drosophila* are these papers on excitatory command neurons that elicit and regulate forward swimming in another arthropod:a) Wiersma CAG, and Ikeda K (1964) Interneurons commanding swimmeret movements in the crayfish, Procambarus clarkii. Comp.Biochem.Physiol. 12:509-525.b) Stein PSG (1971) Intersegmental coordination of swimmeret motor neuron activity in crayfish. J. Neurophysiol. 34:310-318.c) Acevedo LD, Hall WM, and Mulloney B (1994) Proctolin and excitation of the crayfish swimmeret system. J. Comp. Neurol. 345:612-627.

4) In addition, it would be good to add in a section/figure on the other known interneurons in the circuit (e.g. GVLI, PMSI from Nose lab) and how they might fit into the GDL circuit.

---

## [Author Response]

Essential revisions:

*The reviewers were unified in thinking that the data presented were important and largely complete. The major suggestion for improvement of the paper is to place this work more carefully into the constellation of knowledge about coordinating circuits in arthropods so readers will be able to appreciate what the authors have accomplished. 1) Introduction, second paragraph et seq.: "A complete wiring diagram with synaptic resolution…" – good, and true. This goal has recently been achieved, using different methods, and published in these papers:*

a) Smarandache CR, Hall WM, and Mulloney B (2009) Coordination of rhythmic motor activity by gradients of synaptic strength in a neural circuit that couples modular neural oscillators. J. Neurosci. 29:9351-9360.b) Smarandache-Wellmann CR, and Grätsch S (2014) Mechanisms of coordination in distributed neural circuits: Encoding coordinating information. J. Neurosci. 34:5627-5639.c) Smarandache-Wellmann CR, Weller C, and Mulloney B (2014) Mechanisms of coordination in distributed neural circuits: Decoding and integration of coordinating information. J. Neurosci. 34:793-803.

We thank the reviewers for pointing us to the very relevant work performed in crayfish. Indeed, the electrophysiological studies in crayfish produced very significant findings for the understanding of coupled oscillators along sequences of segments. We have added references to this work in the Introduction where we contextualize our work with the current knowledge in the field.

*2) The significance of this coordinating circuit's structure for effective locomotion has been analyzed quantitatively in: Zhang C, Guy RD, Mulloney B, Zhang Q, and Lewis TJ (2014) The neural mechanism of optimal limb coordination in crustacean swimming. PNAS 111:13840-13845.*

We thank the reviewers for pointing out the very interesting quantitative analysis and modeling of the crayfish swimmerets as described in Zhang et al. (2014). Both the crayfish and the *Drosophila* larva employ a mode of locomotion that utilizes forward waves of motor neuron commands. In our manuscript we report on how the contraction of the current segment induces the relaxation of the anterior segment in forward peristalsis, with potentially the bulk of the forward excitation being provided by proprioceptive feedback as described elsewhere. On the other hand, the model shown in Zhang et al. (2014) reports on a network where the oscillator module of the CPG responsible for the swimmeret’s return strokes in each segment elicits the return strokes of the segment anterior to it, which is permitted only when the oscillator for the power strokes in that anterior segment ceases to inhibit the oscillator for the return strokes. While fascinating, we were unable to draw specific parallels between the crayfish oscillator-based abstract model and the EM-reconstructed larval circuit regarding motor wave propagation. Interestingly, the circuit diagrams in the crayfish, in particular the coordination between the two CPGs for power strokes and return strokes perhaps are reminiscent of the coordination between longitudinal and transverse muscles in *Drosophila* larva as described by Heckscher et al. 2012, which falls beyond the scope of our manuscript.

*3) Discussion, last paragraph: “descending neurons from the SEZ […] motif has been modelled in the leech, where…" – interesting observation, and a fair reference. More immediately relevant to work in Drosophila are these papers on excitatory command neurons that elicit and regulate forward swimming in another arthropod:a) Wiersma CAG, and Ikeda K (1964) Interneurons commanding swimmeret movements in the crayfish, Procambarus clarkii. Comp.Biochem.Physiol. 12:509-525.b) Stein PSG (1971) Intersegmental coordination of swimmeret motor neuron activity in crayfish. J. Neurophysiol. 34:310-318.c) Acevedo LD, Hall WM, and Mulloney B (1994) Proctolin and excitation of the crayfish swimmeret system. J. Comp. Neurol. 345:612-627.*

Indeed, we had overlooked that. As any evolutionary neurobiologist would expect, all arthropods will present similar segmental premotor circuitry. We thank the reviewers for highlighting this study on crayfish where, similarly to, those in the leech that we refer to, the role of descending neurons on speed regulation has been elucidated with electrophysiology and modeling. We have addressed this by inserting the corresponding references in the last paragraph of the Discussion, further strengthening our interpretation of the role of descending neurons that contact the excitatory premotor interneurons in the *Drosophila* larva, for which we are thankful.

4) In addition, it would be good to add in a section/figure on the other known interneurons in the circuit (e.g. GVLI, PMSI from Nose lab) and how they might fit into the GDL circuit.

Indeed, while writing this manuscript, several other manuscripts emerged that contributed further neurons to the larval circuits from EM, which enabled us to identify a number of relations with the circuits reported here and prior work from the Nose lab. We composed an additional supplemental figure where we show the EM-reconstructed circuits that relate the inhibitory GDL neurons introduced here with the PMSIs (Kohsaka et al. 2014), the GVLIs (Itakura et al. 2015), and the Even-Skipped+ interneurons that maintain left-right symmetric muscle contraction amplitude (Heckscher et al. 2015), along with the relation of proprioception with these neurons (e.g. Schneider-Mizell et al., in press). We added a new paragraph and figure (Figure 8—figure supplement 2) to the discussion establishing all these relationships.